# Disparities between sustainability of country-level seafood production and consumption

Kayla M. Blincow[1,2]*, Alan C. Haynie[3,4], Brice X. Semmens[2]

**1** College of Science and Mathematics, University of the Virgin Islands, St. Thomas, U.S. Virgin Islands, United States of America, **2** Scripps Institution of Oceanography, University of California, San Diego, La Jolla, California, United States of America, **3** NOAA Fisheries, Alaska Fisheries Science Center, Seattle, WA, United States of America, **4** ICES Secretariat, Copenhagen V, Copenhagen, Denmark

* kblincow@ucsd.edu

## Abstract

Ensuring the sustainability of seafood is increasingly important for supporting food security and stable livelihoods in the face of a growing human population. A country's seafood sustainability is often defined by how it manages its fisheries stocks; however, growth in the volume and complexity of global seafood trade has created an increasing disconnect between the sustainability of a country's seafood production and their seafood consumption. Using a global seafood trade database, we demonstrate wide differences between country-specific sustainability of produced versus consumed seafood. Our findings suggest that countries most consistently considered leaders in seafood production sustainability will inevitably consume seafood held to lower sustainability standards through their participation in the global seafood trade network. This issue is exacerbated by an inability to accurately trace the origins of seafood products based on current international trade reporting practices. Our analysis suggests that improved international seafood tracing and sustainability metrics that account for both production and consumption are critical to future efforts to manage global seafood sustainability.

## 1. Introduction

Freshwater and marine products sourced from wild capture and aquaculture fisheries, referred to from here onward as seafood, play a critical role in global food systems. In 2017, approximately 3.3 billion people derived 20% of their animal protein intake from fish, with this percentage being even higher for many developing and small island nations [1]. In addition to providing sustenance, seafood and fishing industries are an important source of jobs and income for many people around the world, supporting the livelihoods of more than 10% of the global population [1,2]. As demand for fisheries as a source of food and livelihoods increases, so do potential negative environmental impacts stemming from unsustainable fishing (e.g. loss of biodiversity, fisheries-induced evolution through artificial selection for biological traits due to fishing practices, and altered trophic dynamics) [3–5].

**Data Availability Statement:** All code for the analyses performed and raw and intermediate (where necessary) data files can be found at https://zenodo.org/records/13799970. The FAO

Food Balance Sheet data is publicly available via FAO and their FishStatJ Program (https://www.fao.org/fishery/en/topic/166235). The FMI data is available via (Melnychuk et al. 2017; DOI: https://doi.org/10.1038/s41893-020-00668-1) and the Ocean Health Index (https://github.com/OHI-Science/ohiprep/releases). The Global Trade Atlas Dataset is a subscription-based compilation of data provided by Global Trade Information Services, Inc (now S&P Global). Data cannot be shared publicly because it is owned by S&P Global. The data used in this analysis were made accessible by funding from the NOAA Fisheries Office of Science and Technology Economics Program through NOAA's Alaska Fisheries Science Center (AFSC). The data underlying the results presented in the study are available from S&P Global. AFSC Data Contact: Brian Garber-Yonts, Ph.D. Brian.Garber-Yonts@noaa.gov S&P Global Data Contact: Charlie Campbell Account Manager – Global Intelligence & Analytics S&P Global Market Intelligence New York charlie.campbell@spglobal.com charlie.campbell@ihsmarkit.com spglobal.com/marketintelligence

**Funding:** NOAA Fisheries Office of Science and Technology Economics Program funded the organization of the GTA data. Edna Bailey Sussman Fund Graduate Environmental Internship and the NOAA National Marine Fisheries Service Quantitative Ecology and Socioeconomics Training (QUEST) program supported the first author to complete this work.

**Competing interests:** That authors have declared that no competing interests exist.

In recognition of the importance of marine and aquatic resources, such as fisheries, to global development the United Nations (UN) included them in their Sustainable Development Goals for 2030, calling on the global community to "conserve and sustainably use the oceans, seas, and marine resources for sustainable development" (UN General Assembly 2015, Ovando et al. 2021). While international agreements, such as the UN Sustainable Development Goals [6], UN Convention on Biological Diversity [7], UN Convention on the Law of the Sea [8], and UN Food and Agriculture Organization (FAO) Code of Conduct for Responsible Fisheries [9], provide guidelines for sustainability, implementation of these agreements is ultimately left to national governments and management agencies. As a result, global seafood sustainability is highly dependent on national-level fisheries governance.

Sustainable fisheries can be defined in many ways. On a fishery-by-fishery basis in wild capture fisheries, managers often rely on standard indicators to assess and regulate sustainable production, such as maximum sustainable yield (MSY), abundance ($B/B_{MSY}$), or fishing mortality ($U/U_{MSY}$) [10,11]. The effectiveness of these indicators (and thus fisheries management) is dependent on the quality of data available, the usage of appropriate modeling methods, and ultimately, the successful application of regulatory efforts [12]. As such, one measure of fisheries sustainability on the scale of countries is the relative level of fisheries management and enforcement. Increased fisheries management intensity is associated with more sustainable fisheries production [11,13,14]. That is, countries that are better equipped to establish strong fisheries management, particularly in reference to enforcement, fishing regulations, and capacity to conduct stock assessments [14], have relatively few overfished stocks. Similarly, within aquaculture, countries that have the stability and capacity to support strong property rights and establish regulatory oversight of the industry have more strongly managed and sustainable aquaculture production [15]. While countries with stronger capacity to manage their fisheries generally produce sustainable seafood, the sustainability of their consumption is the product of both locally produced seafood and imports from other countries—the latter operating outside the bounds of national fisheries management.

Trade of seafood products is increasingly globalized [16–18]. As some of the most traded food commodities in the world, 78% of seafood products experience competition from international trade and 38% of all fisheries production enters international trade markets [1]. The globalization of seafood markets means that countries are not just consuming seafood products that they produce themselves, but rather are a part of a vast network of international seafood trade that derives products from many different sources. This creates potential for a mismatch between seafood production and seafood consumption sustainability [18–20].

To explore the nature of seafood sustainability, we drew on the well-established relationship between strong fisheries management and increased fisheries production sustainability [11,13,14,20], using a metric of fisheries management intensity, the Fisheries Management Index (FMI), as a proxy for seafood production sustainability. This metric ranges from 0 to 1 and constitutes a weighted average of evaluations by experts of national fisheries management systems based on attributes including research, management, enforcement, and socioeconomics [14]. Linking this measure of seafood production sustainability with a high-resolution seafood trade database, the Global Trade Atlas (GTA), we build on previous studies addressing issues in seafood sustainability (e.g. 18–20) by characterizing the disparities in the sustainability of seafood production and consumption within and across countries.

## 2. Methods

We used three sources of data in our analysis: the FAO Food Balance Sheet of fish and fishery products [21], the GTA dataset [22], and the FMI [14,23]. We used information from these

sources and various assumptions (defined below) regarding the nature of trade of seafood products and consumption to estimate the management intensity of seafood products that are consumed within a country. Using these values we explored seafood sustainability, using the management of seafood products as a proxy to compare sustainability of seafood consumed and produced across countries globally. We performed all analyses using R Statistical Software, version 4.4.1 [24].

## 2.1 Data

The FAO Food Balance Sheet includes information on the production, imports, exports, and total food supply of seafood products (i.e. supply available for consumption) associated with countries grouped by coarse product types and year [21]. The product types in the FAO data are derived from the International Standard Statistical Classification of Aquatic Animals and Plants (ISSCAAP) and include: Freshwater and Diadromous Fish, Pelagic Fish, Demersal Fish, Marine Fish Not Elsewhere Indicated (NEI), Crustaceans, Cephalopods, Molluscs Excluding Cephalopods, and Aquatic Animals NEI. We relied chiefly on the FAO dataset as the core of our analysis due to it being the standard data used in global seafood trade analyses and a lack of a better alternative that includes estimates of the components of seafood consumption across a wide array of countries.

The GTA data include information on the quantity of imports and exports of seafood products (Harmonized System Codes (HS) -03, -16) between trade partners. We will refer to countries that report to the dataset as "reporters" and their associated trade partners as "partners". We used this dataset to define which trade partners contributed to the imports of reporters recorded in the FAO Food Balance Sheet. We filtered out products that are not intended for human food consumption as defined by the FAO Food Balance Sheet metadata, including ornamentals, oils, feed, and capsules. We also filtered out trade relationships that did not apply to our analysis, for example, some countries reported trade that occurred between regions within their national boundaries. Additionally, we filtered out reporters that were grouped or were duplicates of other reporters (e.g. European Union reporters that were aggregates of member countries). The GTA data on imports and exports are recorded in product weight, while the FAO data are recorded in live weight. To combine information from both datasets we converted the product weights in the GTA data to live weights using the suggested conversion factors (CF) from the FAO Coordinating Working Party on Fishery Statistics (CWP) Handbook of Fishery Statistical Standards detailed in Annex I.1 (S1 Table) [25]. In cases where products did not have an FAO CF, we used the most closely taxonomically related CF available. There were some cases (0.2% of total imports) that did not have a close equivalent FAO CF (e.g. caviar and shark fins) or did not have information regarding the product (e.g. confidential product trade). In these cases, we used the best available CF from other sources or assigned a CF of 1 (see S1 Table). We chose to assign a CF of 1 to product groups with no information, such as confidential trade, as we did not have a basis for determining what an appropriate CF would be other than the product weight itself. Additionally, we classified all GTA products into the FAO ISSCAAP product groupings used in the FAO Food Balance Sheet data.

The FMI data constitute estimates of the management intensity of wild capture fisheries in different countries. Originally created by Melnychuk et al. (2017), FMI is a metric ranging from 0 to 1 that uses information from surveys of fisheries experts to assess research, management, enforcement, and socioeconomics of fisheries management across 28 different countries that account for > 80% of global catch. Melnychuk et al. later expanded the FMI to include additional countries and fisheries stocks [11,26]. The Ocean Health Index (OHI) estimated FMI for

additional countries/territories to incorporate it into their resilience estimates of coastal countries for their 2019 Ocean Health Index global assessment [23,27]. To do so, they used linear models to determine which variables best predicted FMI from an original set of 40 countries from Melnychuk et al., including various measures of gross domestic product (GDP), governance, and region. They found that the Social Progress Index (SPI) and UN geo-regions were the best predictors of FMI. SPI is an index that measures how well countries support basic human needs, foundations of wellbeing, and opportunity for their citizens [28]. After fitting a linear model using these explanatory variables, they used it to fill gaps in the FMI dataset to include 80 additional countries/territories [23]. There were a total of 324 different trade partners reported in the GTA dataset, some of which represented smaller territories within countries or groupings of multiple countries (S2 Table). For trade partners that constituted groups of countries (e.g. European Union) or unrecognized territories we determined FMI by either calculating the mean FMI if it was a group of countries or finding the most closely related country or territory if it was a single unaccounted for trade partner. For the remaining countries (n = 44), we used the OHI gap filling model to further expand the FMI dataset.

While FMI was created as a measure of management intensity for wild capture fisheries, we applied it to both wild capture and aquaculture seafood products in our core analysis since the trade datasets do not specify the method of production. We believe this to be a reasonable assumption considering many of the country-level factors influencing management intensity of wild capture seafood products (e.g. enforcement capacity) have similar influences on the management of aquaculture seafood products [15]. However, we also conducted a parallel analysis where we removed products potentially associated with aquaculture from the GTA data, essentially removing trade relationships that are based solely on aquaculture products, and then compared these results to our original analysis. We will refer to this analysis as the aquaculture exclusion analysis from this point onward. We chose to account for aquaculture in this way, because the coarseness of FAO product groupings is such that it is impossible to unequivocally distinguish products derived from aquaculture. To identify products potentially associated with aquaculture, we first removed any products explicitly classified as "farmed", "cultured", or "cultivated" from the dataset. Next, we removed any trade of GTA products associated with species listed in Table 10 (World Production of Major Aquaculture Species; Including Species Groups) of the FAO World Review on The State of World Fisheries and Aquaculture 2022 Report (30), unless the products were explicitly classified as "not farmed" or "excluding farmed". Finally, we excluded generalized product listings associated with common aquaculture species (e.g. oysters, shrimp, etc.) as there was no way to tell their mode of capture.

Given the strength of the positive relationship between the level of fisheries management and sustainable production of seafood [11,14], we characterized FMI as a metric of seafood sustainability. For the purposes of our analysis, we refer to the FMI data of country-level management intensity as $FMI_P$, or the sustainability of a country's seafood production. These data also formed the basis for our calculation of $FMI_I$, or the sustainability of a country's imports of seafood products, and $FMI_C$, or the sustainability of a country's consumption of seafood products (see below).

## 2.2 Calculating $FMI_C$

We combined the three datasets described above to determine the FMI associated with consumption ($FMI_C$) for each reporter present in both the GTA and FAO datasets. The GTA data did not have high resolution trade information for all reporters prior to 2012, and the FAO dataset only included data up to 2017. For these reasons we limited our analysis to the years 2012–2017.

We used the same equation for calculating consumption that underpins the FAO Food Balance Sheet calculation of total food supply to serve as the basis for our calculations (Eq 1). This equation states that the product that is available for human consumption is what is left over after accounting for inputs of product from domestic production and imports from other countries, and outputs of product associated with exports:

$$C = P + I - E \tag{1}$$

For any given reporter and product group, C is consumption of the seafood product, P is domestic production of the product, I is imports of the product from other countries, and E is exports of the product to other countries. Building from this fundamental relationship we generated an equation for calculating $FMI_C$ for each country that scales FMI based on the relative contributions of product available for consumption from domestic production and imports (Eq 2):

$$FMI_{C_r} = \left( p_{P_{Cr}} \times FMI_{P_r} \right) + \left( p_{I_{Cr}} \times FMI_{I_r} \right) \tag{2}$$

For reporting country $r$, $p_{P_{Cr}}$ is the proportion of consumption of a product derived from domestic production, $p_{I_{Cr}}$ is the proportion of consumption derived from imports from other countries, $FMI_{P_r}$ is the production FMI for the reporting country, and $FMI_{I_r}$ is the FMI associated with the trade partners supplying imports of the product scaled based on their contribution to total imports to country $r$. We calculated $FMI_{I_r}$ as follows (Eq 3):

$$FMI_{I_r} = \sum_{j=Partner1}^{n} \left( \left( p_{P_{Ej}} \times p_{I_{rj}} \times FMI_{P_j} \right) + \left( p_{I_{Ej}} \times p_{I_{rj}} \times FMI_{glbl} \right) \right) \tag{3}$$

Where $p_{P_{Ej}}$ is the proportion of exports of a product by trade partner $j$ derived from their domestic production, $p_{I_{Ej}}$ is the proportion of exports by trade partner $j$ derived from their imports, $p_{I_{rj}}$ is the proportion of imports of the reporting country $r$ derived from trade partner $j$, $FMI_{P_j}$ is the production FMI for trade partner $j$, and $FMI_{glbl}$ is a global estimate of the FMI for a given product scaled based on the relative production by countries (Eq 4):

$$FMI_{glbl} = \sum_{k=country1}^{n} \left( p_{Pglbl_k} \times FMI_{P_k} \right) \tag{4}$$

Where $p_{Pglbl_k}$ is the proportion of global production of a product associated with country $k$, and $FMI_{P_k}$ is the production FMI associated with country $k$. There were some instances where we could not directly calculate $p_{Pglbl_k}$. This largely occurred in instances where the GTA trade partners did not align with the FAO countries (e.g. grouped EU trade relationships). In those instances, we decided the production proportion on a case-by-case basis, usually taking the mean for countries included in the group (see S3 Table).

## 2.3 Assumptions and sensitivity of unknown parameters

In the equations above, there are 3 key unknowns that we were unable to estimate with the data in hand: 1) the proportion of consumption of a product that is derived from domestic production ($p_{P_{Cr}}$), 2) the proportion of exports of a product that is derived from domestic production ($p_{P_{Ej}}$), and 3) country of origin for any re-exported seafood products. Each of these unknowns are the result of a common, and poorly documented practice: re-exporting and/or re-importing products. While some countries do report re-exports, the recognized definition of re-exports within global trade only includes products that are imported and re-exported in

the same form. Products that are imported and undergo processing get reported as new products originating from the point of processing. For example, if country A imports whole fish from country B, then processes and re-exports that product as canned fish, that fish becomes a new product attributed to country A. As a result, tracing the origin of seafood products that undergo processing in countries separate from where they are harvested is currently not possible with the available data. We refer to this challenge as the issue of re-exports. Since our goal was to trace the origin of seafood products from their original country of raw production (capture), we were required to make assumptions regarding these values. To address these unknowns, we conducted a parameter sensitivity analysis based on prior parameter estimates in the literature.

Regarding the proportion of consumed seafood products that are domestic in origin, we used a range of parameters, including a naïve assumption and two values derived from the literature. First, we calculated $FMI_C$ under the assumption that, for all countries and products, consumption that was domestic in origin was proportional to the amount of domestic production of that product. For example, if a country produced 50 tons of fish and imported 10 tons of fish, we assumed the total food supply available for consumption derived from domestic production versus imports was 5:1 ($p_{P_{Cr}} : p_{I_{Cr}}$). We refer to this $FMI_C$ estimate as the proportional derivation. Second, we relied on an estimate of the proportion of consumption associated with domestic production made in Gephart et al. (2019). They estimated that 35–38% of seafood consumption in the United States of America (USA) is produced domestically, accounting as best as possible for the issue of re-exports. We applied a $p_{P_{Cr}}$ of 0.365 to all countries for our second estimate of $FMI_C$, which we refer to as the Gephart derivation. We recognized that it is not reasonable to assume all countries would operate under similar circumstances to the USA regarding seafood trade dynamics, but this was the most carefully developed existing estimate. Our final estimate used a value from Guillen et al. (2018) that used a Multi-Region Input-Output model (MRIO) to characterize global seafood consumption and estimated that approximately 74% of final consumption of seafood products is from domestic supply globally [19]. Their MRIO model did not directly account for the issue of re-exports, and assumed products originated from the country of export. We applied a $p_{P_{Cr}}$ of 0.74 to all countries for our third estimate of $FMI_C$, which we refer to as the Guillen derivation.

To calculate $FMI_I$ we needed to make assumptions regarding the proportion of imports from each partner that were derived from the partner's domestic production versus re-exports from previously imported product ($p_{P_{Ej}}$ and $p_{I_{Ej}}$; Eq 3). We used the assumption that both $p_{P_{Ej}}$ and $p_{I_{Ej}}$ were proportional to the partner country's amount of production and imports of the given product. This assumption allows us to simultaneously leverage the power of the highly resolved bilateral trade flows present in the GTA data while still allowing our results to correct for the issue of re-exports by attributing a conservative proportion of exports to a global mean FMI (described below).

Finally, regarding the country of origin for any re-exported seafood products, we made the regularizing assumption that re-exported products had an FMI equivalent to the global average of the FMI for all countries for each product group. We scaled this global estimate based on the level of production of each product by each country (Eq 4).

## 2.4 Analysis

We calculated all the equations above based on the FAO product groups and aggregated the final $FMI_C$ estimates based on the relative proportion of total seafood consumption associated with each product. All the $FMI_P$ values were drawn from the FMI dataset. The GTA data

provided the relative contribution of different trade partners to each reporter's imports. The FAO data provided information on the total food supply and relative proportions of different product groups to the total food supply for each reporter. We compared the total live weight of imports by country from the GTA dataset against the FAO dataset to determine how much they differed, and found that the FAO data had lower estimates, in some cases substantially so (see S1 Fig). Despite the disparities in the magnitude of trade reported by the different datasets, we assumed that the relative contribution of trade partners derived from the GTA data was the same for the FAO data.

We analyzed our resulting estimates of $FMI_C$ using direct comparison, linear modelling techniques, and calculating the percent change. We used simple linear models to determine the relationship between $FMI_P$ and our estimates of $FMI_C$. We calculated the percent change between $FMI_P$ and $FMI_C$ using the equation:

$$\text{Percent Change} = \left( \left( FMI_p - \text{mean}(FMI_C)/FMI_p \right)^* 100 \right) \tag{5}$$

We further explored the trade relationships associated with the largest importers and exporters globally by looking at the relative contributions and differences in $FMI_P$ among their trade partners.

## 3. Results

Countries associated with more sustainable seafood production, on average, consume seafood products at lower sustainability levels than what they produce (Figs 1 and S2). Those countries at the upper end of seafood production sustainability, which are known for maintaining well-managed fisheries stocks [11,29,30], will inevitably see decreases in the sustainability of their seafood consumption the more they rely on imports from other countries. We found that the percent change between production and consumption sustainability for the most sustainable producers was surprisingly large; for instance, the highest disparities by country included the USA (22.95% decrease from production to consumption sustainability), Canada (13.89%), and the United Kingdom (10.28%) (Table 1). For the aquaculture exclusion analysis, we found the highest disparities among the USA (23.43%), New Zealand (14.48%), Canda (12.02%), and the United Kingdom (10.54%) (S4 Table). We also saw a shift in the top 25 seafood producers, with Argentina and India no longer making the top 25 and New Zealand and South Africa being added (S4 Table).

Overall $FMI_C$ had a significant positive linear relationship with $FMI_P$, which was expected given the $FMI_C$ calculations were dependent on $FMI_P$ (Tables 2 and S5). The estimates of the linear coefficients varied across the different $FMI_C$ derivations (Figs 2 and S3). The Guillen $FMI_C$ derivation had the highest slope estimate and was the closest to a 1:1 relationship between $FMI_P$ and $FMI_C$ ($FMI_C = 0.797 \times FMI_P + 0.114$; $R^2 = 0.98$), followed by the proportional estimate ($FMI_C = 0.611 \times FMI_P + 0.206$; $R^2 = 0.80$), then the Gephart estimate ($FMI_C = 0.512 \times FMI_P + 0.293$; $R^2 = 0.79$) (Fig 2). Countries with $FMI_P$ values at the extreme ends of the spectrum differed the most from the 1:1 relationship (Figs 2 and S3). In the aquaculture exclusion analysis, the relative relationships were the same, with minimal differences in the slope estimates (S5 Table).

The top five exporters by volume across 2012 to 2017 in descending order were China, Norway, Thailand, Russia, and the USA (Fig 3A). The top five importers in descending order were the USA, China, Japan, Spain, and France (Fig 3B). The highest exporters tended to trade with more partners than the highest importers (Fig 3). There was no difference in these results

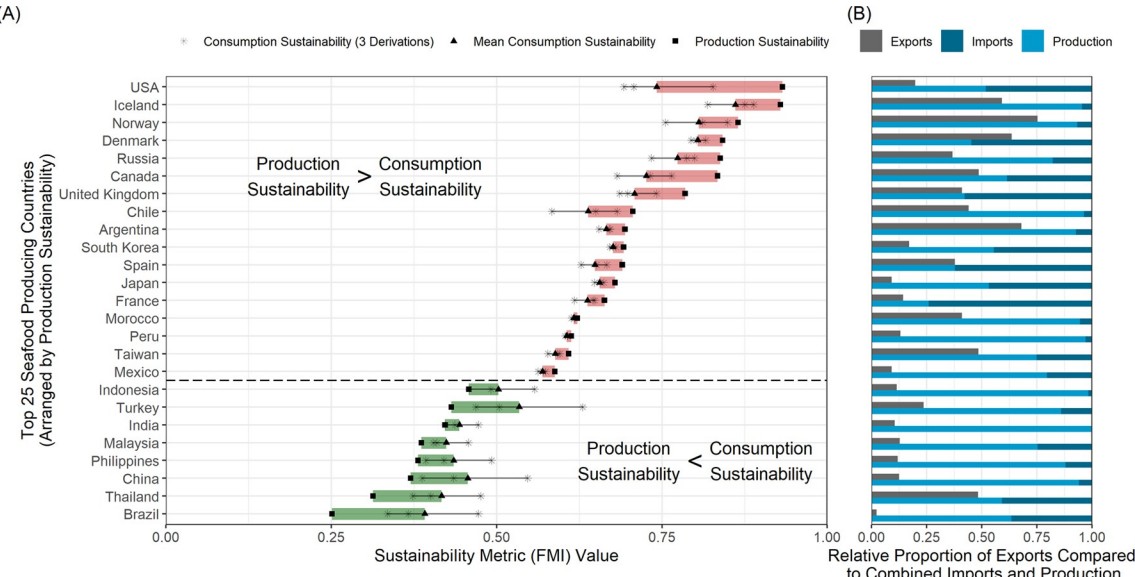

**Fig 1. Plot showing disparity in production and consumption sustainability and the relative proportions of imports, exports, and production for the top 25 seafood producing countries from 2012–2017.** (A) Depiction of the difference between production and consumption sustainability as measured by the Fisheries Management Index (FMI). The colored bars show the difference between the production sustainability (FMI_P—squares) and the mean of the consumption sustainability derivations (FMI_C—triangles), with red indicating a decrease in consumption sustainability compared to production sustainability and green indicating an increase. The gray lines show the range, and the asterisks show the direct estimates of each of the consumption sustainability derivations (FMI_C−Proportional, Guillen, and Gephart). Countries that produce more sustainable seafood than they are consuming occur above the dotted line, while countries that produce less sustainable seafood than they are consuming occur below. (B) The second panel displays the relative proportion of exports (gray bar) compared to combined total imports (dark blue bar) and production (light blue bar) for each country. Note that we do not know what proportion of exports is made up of production versus imports, and thereby do not know what proportions of production and imports are left for consumption. This accounts for the uncertainty in the consumption sustainability estimates in the left panel.

under the aquaculture exclusion analysis, though the magnitude of imports and exports shifted slightly (S4 Fig).

China and the USA contributed most to global exports and imports, respectively (Figs 3 and S4). The USA traded largely with countries that have lower production sustainability (Figs 4A and S5). Most imports coming into the USA were from Asian countries (>57% of total imports), with the largest contributions from the region coming from China (24.8%), Vietnam (9.4%), Thailand (6.7%), and Indonesia (6.3%) (Fig 4A). The second largest contributor to USA imports after China was Canada (13.9%) (Fig 4A). China's exports are traded globally, with the largest proportions of its exports going to Japan (17.9%), South Korea (15.1%), and the USA (14.6%) (Fig 4B). A majority of China's major trade partners had higher production sustainability than China itself (0.370; Fig 4B). The aquaculture exclusion analysis found the same overall relationships (S5 Fig). Most USA imports were from Asian countries (>50% of total imports), with the largest contributions coming from China (21.3%), Vietnam (12.5%), Thailand (6.8%), and Indonesia (5.6%) (S5 Fig). The second largest contributor to USA imports after China was still Canada (16.3%) (S5 Fig). In the aquaculture exclusion analysis, the countries contributing most to China's exports were still Japan (18.1%), South Korea (18.1%), and the USA (10.8%) (S5 Fig).

The aquaculture exclusion analysis did not deviate substantially from the results of the core analysis (see supplementary material). While the magnitude of the results differed slightly, the trends observed across all analyses were robust to the sustainability assumptions associated with aquaculture products.

**Table 1. Percent change in production and consumption sustainability for the top 25 seafood producing countries (according to FAO data).** Production sustainability is the $FMI_P$ for each country, and mean consumption sustainability is the mean of the three $FMI_C$ derivations. We calculated the percent change between these values for each country by subtracting the mean consumption sustainability from the production sustainability and dividing by the production sustainability and multiplying by 100 (($FMI_P$−mean($FMI_C$)/$FMI_P$)*100). The nature of this equation means that positive percent change values occur for countries that have a decrease in consumption sustainability compared to production sustainability and negative values occur for countries that have an increase.

| Country | Production Sustainability ($FMI_P$) | Mean Consumption Sustainability mean($FMI_C$) | Percent Change (%) |
|---|---|---|---|
| United States of America | 0.932 | 0.742 | 22.95 |
| Iceland | 0.929 | 0.861 | 7.66 |
| Norway | 0.865 | 0.805 | 7.25 |
| Denmark | 0.842 | 0.804 | 4.57 |
| Russia | 0.838 | 0.774 | 8.05 |
| Canada | 0.834 | 0.726 | 13.89 |
| United Kingdom | 0.785 | 0.709 | 10.28 |
| Chile | 0.706 | 0.638 | 10.23 |
| Argentina | 0.694 | 0.666 | 4.17 |
| South Korea | 0.692 | 0.676 | 2.37 |
| Spain | 0.690 | 0.647 | 6.27 |
| Japan | 0.679 | 0.655 | 3.53 |
| France | 0.663 | 0.637 | 3.98 |
| Morocco | 0.622 | 0.616 | 0.89 |
| Peru | 0.613 | 0.591 | 1.15 |
| Taiwan | 0.609 | 0.588 | 3.43 |
| Mexico | 0.588 | 0.569 | 3.29 |
| Indonesia | 0.458 | 0.502 | -8.90 |
| Turkey | 0.431 | 0.534 | -20.40 |
| India | 0.422 | 0.442 | -4.93 |
| Malaysia | 0.386 | 0.423 | -9.13 |
| Philippines | 0.381 | 0.435 | -12.78 |
| China | 0.370 | 0.452 | -19.73 |
| Thailand | 0.313 | 0.416 | -27.79 |
| Brazil | 0.251 | 0.391 | -42.42 |

## 4. Discussion

Countries that are known for producing well-managed, sustainable seafood are leaning heavily on countries with less sustainable management practices to supply the seafood they consume. Our findings not only highlight the overall disparity in management intensity globally, but also the tendency for countries to trade across this management intensity gradient. Many countries, especially in the developed world, are net importers of seafood, meaning they consume more than they produce [31–33]. It follows that the sustainability of their seafood consumption will shift principally based on the countries from which they import.

**Table 2. Linear model results estimating the relationship between $FMI_P$ and the three $FMI_C$ derivations (Model Equation: $FMI_C \sim FMI_P$).** SE: Standard Error.

| $FMI_C$ Derivation | Intercept estimate | Intercept SE | Slope estimate | Slope SE | Slope t value | p-value | $R^2$ |
|---|---|---|---|---|---|---|---|
| Proportional | 0.206 | 0.023 | 0.611 | 0.034 | 18.039 | <0.0001 | 0.80 |
| Guillen | 0.114 | 0.008 | 0.797 | 0.011 | 69.62 | <0.0001 | 0.98 |
| Gephart | 0.293 | 0.020 | 0.511 | 0.030 | 17.45 | <0.0001 | 0.79 |

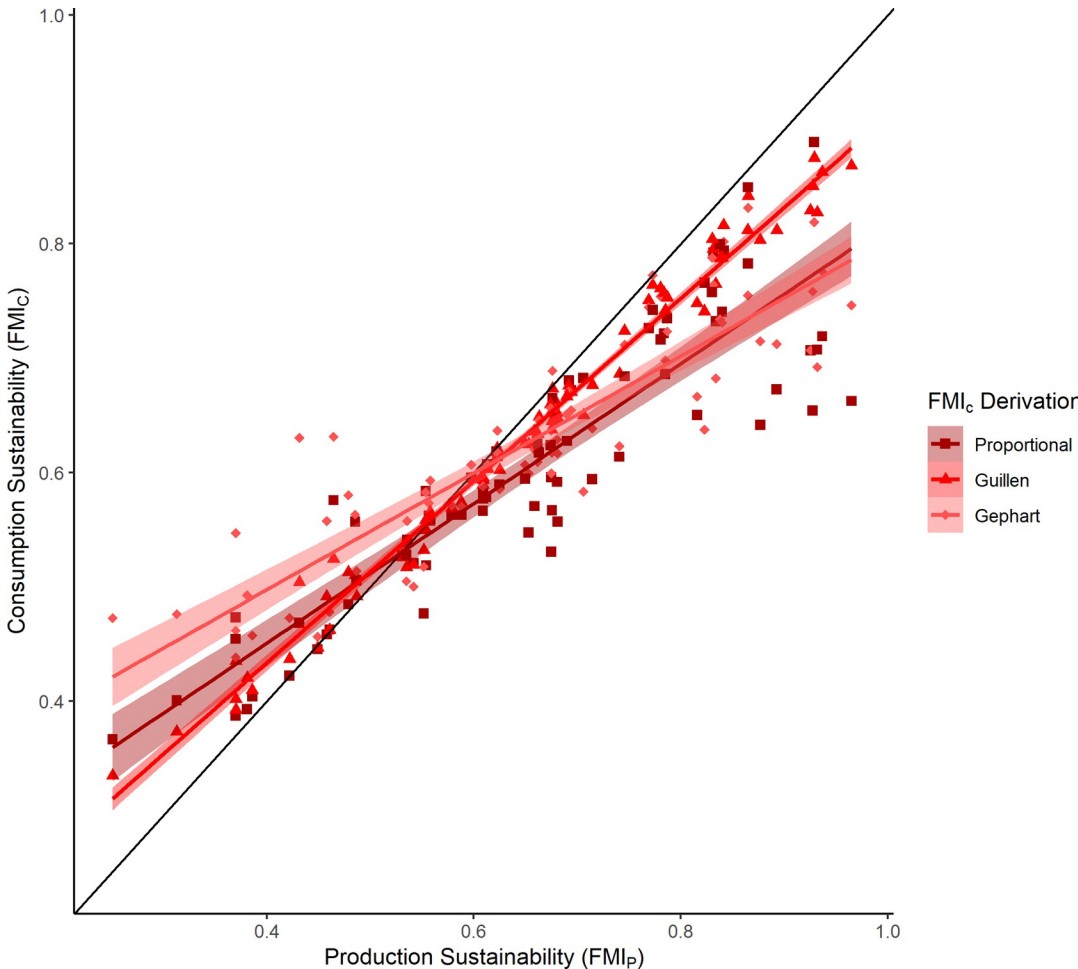

**Fig 2. Plot comparing production to consumption sustainability across all countries analyzed.** The points depict the raw data (countries), while the lines show the linear model results. The shaded areas around the lines denote the 95% confidence intervals for the linear model fit. The different colors denote the different consumption sustainability ($FMI_C$) derivations. The black line shows the hypothetical direct 1:1 relationship between the two variables for comparison.

In general, developed countries, as defined by the UN, import most of their seafood products from developing countries [18,32,33]. Decreased fisheries production in developed countries, potentially because of declines in stock status and more stringent management in recent years, has led to an increased reliance on seafood imports among these countries [34]. An increase in the production and export of seafood from the developing world has largely met this demand [17,34,35]. Fisheries management and governance in the developing world is generally less extensive than in the developed world [3,17,34], and as such this flow of exports from developing to developed countries contributes to a discrepancy in the overall sustainability of seafood consumption in developed countries compared to their production. This relationship is reflected in our analysis results. Furthermore, developing countries tend to import less seafood and rely more heavily on their own production to supply their seafood consumption. For example, the FAO data reports that imports only account for 0.3% of the total seafood inputs into India ($FMI_P$ = 0.422), which is classified as a developing country by the UN [14,36]. As such, in both developed and developing countries, seafood consumption is heavily reliant on products that are subject to less stringent sustainability standards.

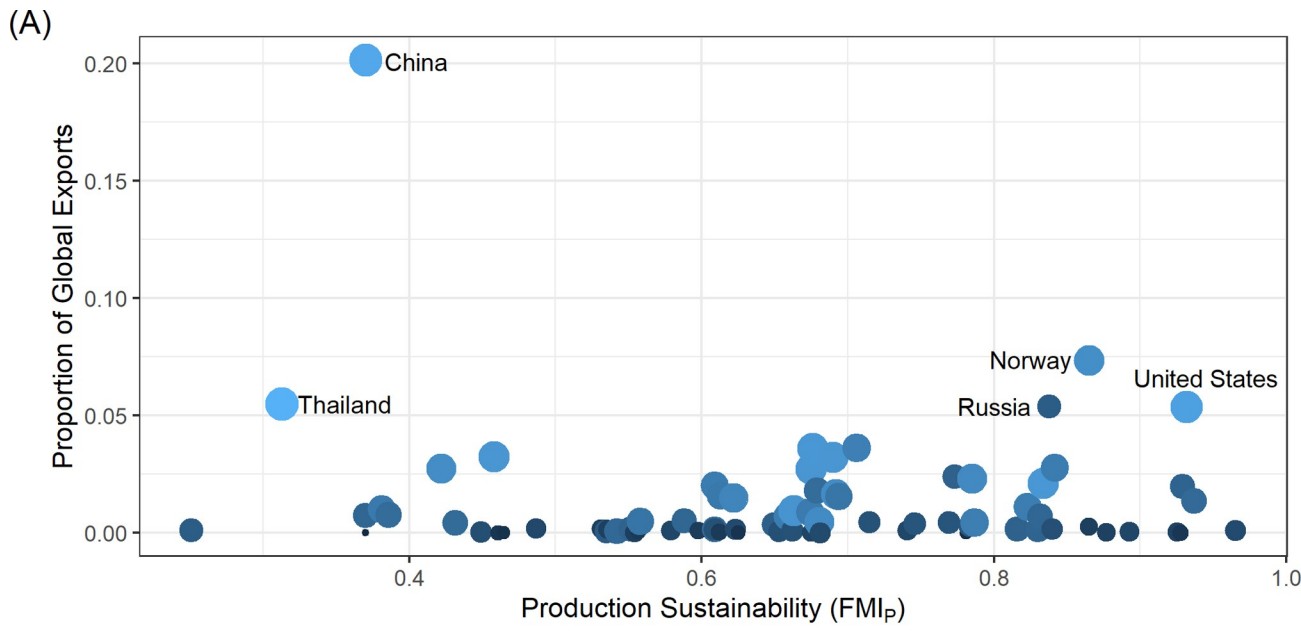

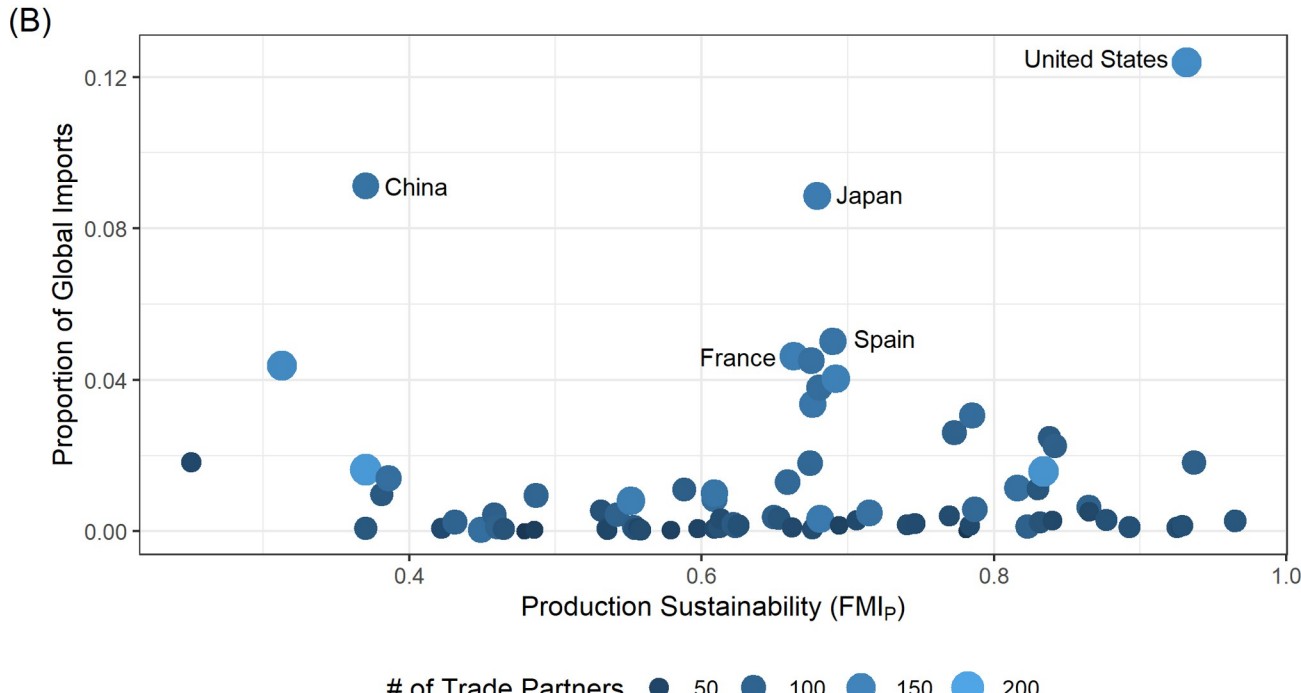

**Fig 3.** The proportion of global exports (A) and imports (B) from 2012–2017 for each country analyzed arranged by $FMI_P$. The size and color of the points shows the number of trade partners for each country (note difference in scale of y-axis). The top five exporters and importers are labeled in each panel.

We can see this relationship clearly in our examination of the USA, the largest importer of seafood globally across the time period we studied. The USA is estimated to rely on imports for 62–65% of their seafood consumption [30]. Every trade partner in the top 25 sources of imports to the USA from 2012–2017 had lower fisheries production sustainability than the USA, which is largely to be expected given that the USA is one of the top producers of

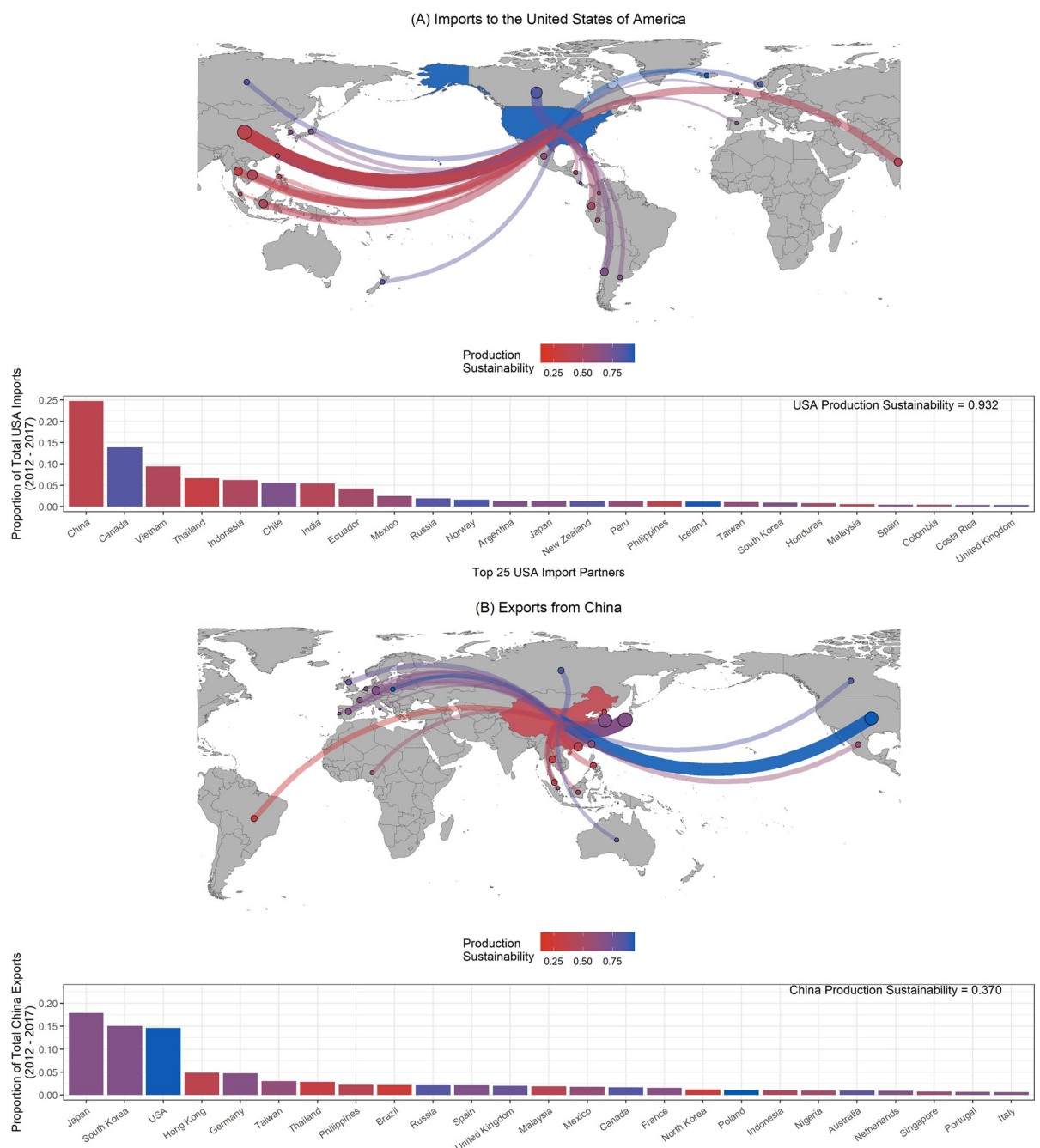

**Fig 4.** Figure depicting (A) imports *to* the United States of America (USA) from their top 25 import partners and (B) exports *from* China to their top 25 export partners from 2012–2017. In the maps, China and the USA are shaded based on their own production sustainability. The lines and points show the origin of imports to the USA (A) and destination of exports from China (B), respectively, with the color denoting the production sustainability of the trade partner. The thickness and darkness of the lines and points are scaled to the magnitude of the trade going to/from each country. The bar plots depict the proportion of total USA imports or Chinese exports attributable to each of top 25 trade partners, respectively. The USA's top 25 import partners account for 94.68% of total imports to the USA from 2012–2017. China's top 25 export partners account for 89.00% of total exports from China from 2012–2017. The bars are shaded based on the trade partner's production sustainability. The base maps were made using public domain data from Natural Earth (naturalearthdata.com).

sustainable seafood. Any country on the upper end of the production sustainability spectrum will see a decrease in their consumption sustainability if they engage in seafood trade; however, most USA imports originated from countries from the lower half of the production sustainability spectrum. Thus, the USA is largely consuming seafood that is held to markedly different management and sustainability standards than what they produce. The strength of this disparity for the USA (core analysis percent change of production and consumption sustainability of 20.36%) is highlighted when you compare them against countries with comparable production sustainability, such as Iceland (7.33%), Norway (6.89%), and Denmark (4.46%), which all also engage in global seafood trade, but display a substantially smaller disparity between the sustainability of their seafood production and consumption. The differences observed between these countries and the USA are closely linked to the geopolitical climate, international trade agreements, local food safety standards and regulations, and culture-driven demand for specific seafood products. For example, the European Union has strict rules on imports of animal products from China, even banning imports of aquatic products completely in the past due to instances of Chinese imports violating local food safety standards [37,38].

From the perspective of the largest exporters of seafood, it is apparent that the observed disparities in sustainability are exacerbated by the dominance of China in the seafood export market. In both of our analyses, China accounted for over 20% of global exports of seafood. This is due to their own substantial seafood production but is also closely linked to their dominant role in global seafood processing [39]. Not only is China responsible for the largest proportion of exports, but those seafood products are distributed with remarkably broad global reach. As a result, the sustainability of seafood produced in China is a part of the seafood consumption sustainability budget of most countries across the globe.

Many seafood products traverse complex global processing and supply chains before they reach their final destination [31,40]. When products are imported, processed, and exported as new products they get attributed to the country that did the processing, not the country that initially harvested them [30,32,41]. As a result of these trade dynamics, like other accounts of global seafood consumption [30,32], we relied on assumptions to estimate the proportion of a country's consumption and exports associated with their own production versus imports. There are some cases where it is likely that the challenges associated with accounting for re-exports resulted in our under- or over-estimating the proportion of seafood consumption associated with domestic versus imported fisheries products. Again, China is the largest seafood exporter in the world, but also one of the largest seafood processors, with an estimated 75% of all seafood imports being processed and re-exported [39]. We handled the lack of transparency in re-exporting of seafood products through sensitivity analyses and are confident that the overall trends we present are robust. Nevertheless, the uncertainty in our findings highlights one of the largest challenges facing the promotion of global seafood sustainability— the inability to accurately account for the fate of seafood products from capture to consumption.

This issue of re-exports is just one in a list of challenges facing seafood traceability. Fisheries reporting largely fails to capture transshipment, or offloading of catch to refrigerated vessels at sea, masking original catch locations [42]. There are also issues with illegal, unreported, and unregulated (IUU) fishing practices [41,43], mislabeling [32], high seas fishing, and private fisheries agreements that result in products being attributed to countries separate from where they are harvested [34,35]. Furthermore, there is no direct reporting of whether products are derived from capture fisheries or aquaculture. While we made an effort in this study, without a reliable means to track products from their source to their consumption, it is challenging to accurately estimate the sustainability of seafood consumption [30,32,39]. Beyond academic

pursuits, this lapse in the accounting of seafood traceability makes efforts to promote sustainability at the consumer level extremely difficult [30,32].

A potential spillover effect of the inability to determine the origin of seafood is a shielding of less sustainable products from economic disincentives to production, and thus an indirect promotion of unsustainable fisheries. That is, while consumers have the power to promote improved fisheries management practices through their seafood purchases [44,45], doing so requires accurate information on seafood sourcing. Certification systems and sustainable seafood guides, such as those created by the Marine Stewardship Council, attempt to leverage this consumer purchasing power by rating the sustainability of seafood products, in part based on the level of fisheries management [45,46]. Even though many of the species included in these programs are assessed differently by region or country of origin, the complex nature of seafood markets and the inability to trace seafood products greatly hinder the efficacy of sustainability certifications and guides [40,47]. We believe the extent to which this shielding of less sustainable products occurs warrants further investigation.

It is tempting to demonize countries, like China, that produce and export large quantities of apparently less sustainable seafood. However, the inability of consumers to determine the sustainability of their seafood creates a market failure in which consumers who are willing to pay for sustainability are unable to reliably purchase sustainable products [48]. This is analogous to the famous 'market for lemons' [49], in which consumers' inability to determine the quality of a product leads to the collapse of the market for the high-value product. Clearer information about sustainable products could end this market failure and increase sustainable production. Unfortunately, the failure of the global trade system to provide consumers with a means to determine the sustainability of product choices takes away the incentive for producers to invest in increasing sustainability. In the incredibly competitive global seafood industry, this market failure makes an aggressive commitment to sustainable production economically perilous, especially for wholesalers that cannot establish brand identity with customers. To increase the sustainability of seafood on a global scale, we argue that we must first address the biggest hurdle present in this study, the opacity of trade reporting.

How can global seafood traceability be improved in the face of growing trade complexity? At the level of trade reporting, Chan et al. (2015) suggest extending the international standardized system of names and codes for trade classification (HS) to include 10 digits that would allow for more specificity in product reporting [50]. While increased specificity in customs product reporting would certainly help decrease the uncertainty in tracing supply chains, there is still a need for a mechanism to track seafood from the point of capture to the point of consumption. Emerging technologies such as blockchain provide one avenue for addressing this need. Blockchain in the context of fisheries would provide a definitive, immutable digital record of the path of a seafood product from the point of capture to the point of consumption [40]. This type of technology is not without barriers to implementation including regulatory uncertainty, limited interoperability, and lack of centralized management [40,51]. Nevertheless, if blockchain technology continues to improve and become less expensive, it could measurably improve seafood traceability and consumption accounting.

Developed countries that have the capacity to do so have largely implemented intensive fisheries management that supports sustainable seafood production [14,20,34]; however, a country's seafood consumption is the result of a complex network of global production and trade. We demonstrated that even the countries with the best fisheries management are likely consuming seafood held to much lower sustainability standards. It follows that these high production sustainability countries are contributing via their seafood consumption to the economic drivers behind the unsustainable fishing practices of countries with less intensive management. Thus, any assessment of national seafood sustainability that does not account

for the role of trade in seafood consumption will continue to paint a skewed picture of sustainability that is particularly rosy for the wealthiest countries.

## Supporting information

**S1 Table. Live weight conversion factors and FAO product groups.** The GTA data is recorded in product weight, while the FAO data is recorded in live weight. To combine information from both datasets it was necessary to convert the product weights to live weights in the GTA data. For shark fins we used an estimate based on previously published studies investigating the fin to body weight ratio of blue sharks (Prionace glauca), one of the most commonly caught species in shark fisheries (1–3). For livers/roes/milts, fish heads/fins/maws, and caviar/caviar substitutes we used the ratios for cod livers/roes suggested by the FAO's Measures, Stowage Rates and Yields of Fishery Products publication (4). Additionally, we categorized each GTA product into the corresponding FAO ISSCAAP group. Below is a list of all products in the GTA data, the live weight conversion factors we used, the motivation behind the conversion factor, and the associated FAO product group.
(PDF)

**S2 Table. Reconciling FMI gap filling data.** The OHI gap filling analysis for estimating FMI values for countries not included in the original analysis by the Melnychuk group did not cover all possible reporters and partners in the GTA dataset. Where possible, we used the same model as OHI to continue gap filling (which relied on the Social Progress Index (SPI) and UN Georegions). In some cases SPI was not available, particularly for subregions of larger countries, groups of countries, or unrecognized territories. Below is a list of region/country equivalency and FMI calculation assumptions we made in order to complete the FMI data matching with the GTA data.
(PDF)

**S3 Table. Proportion of production estimates for trade partners.** Some trade partners did not have an associated proportion of production. This largely happened in instances where the trade partners did not align with the FAO countries (e.g. grouped EU trade relationships). For those instances, we decided the production proportion on a case-by-case basis, usually taking the mean for countries included in the group. Below is a list of all such trade partners, whether the product in question was a fish or invertebrate, and how we derived the estimate of production proportion.
(PDF)

**S4 Table. Percent change in production and consumption sustainability for the top 25 seafood producing countries using the GTA wild capture data (according to FAO data).** Production sustainability is the $FMI_P$ for each country, and mean consumption sustainability is the mean of the three $FMI_C$ derivations. We calculated the percent change between these values for each country by subtracting the mean consumption sustainability from the production sustainability and dividing by the production sustainability and multiplying by 100 (($FMI_P$−mean $(FMI_C)/FMI_P)*100$). The nature of this equation means that positive percent change values occur for countries that have a decrease in consumption sustainability compared to production sustainability and negative values occur for countries that have an increase. *Major differences compared to core analysis*: Notably, the top 25 seafood producing countries changed with the removal of potential aquaculture products. The top 6 countries remained the same. New Zealand and South Africa made the top 25, while Argentina and India are no longer on the list.
(PDF)

**S5 Table. Linear model results estimating the relationship between $FMI_P$ and the three $FMI_C$ derivations using the GTA wild capture data (Model Equation: $FMI_C \sim FMI_P$).** SE: Standard Error.
(PDF)

**S1 Fig. Comparison of import quantities by country from the GTA and FAO datasets.** Each point represents a single country with the line showing the hypothetical one-to-one relationship that would occur if the datasets matched.
(PDF)

**S2 Fig. Aquaculture exclusion analysis: Plot showing disparity in production and consumption sustainability and the relative proportions of imports, exports, and production for the top 25 seafood producing countries from 2012–2017.** (A) Depiction of the difference between production and consumption sustainability as measured by the Fisheries Management Index (FMI). The colored bars show the difference between the production sustainability ($FMI_P$—squares) and the mean of the consumption sustainability derivations ($FMI_C$—triangles), with red indicating a decrease in consumption sustainability compared to production sustainability and green indicating an increase. The gray lines show the range, and the asterisks show the direct estimates of each of the consumption sustainability derivations ($FMI_C$−Proportional, Guillen, and Gephart). Countries that produce more sustainable seafood than they are consuming occur above the dotted line, while countries that produce less sustainable seafood than they are consuming occur below. (B) The second panel displays the relative proportion of exports (gray bar) compared to combined total imports (dark blue bar) and production (light blue bar) for each country. Note that we do not know what proportion of exports is made up of production versus imports, and thereby do not know what proportions of production and imports are left for consumption. This accounts for the uncertainty in the consumption sustainability estimates in the left panel.
(PDF)

**S3 Fig. Aquaculture exclusion analysis: Plot comparing production to consumption sustainability across all countries.** The points depict the raw data (countries), while the lines show the linear model results. The shaded areas around the lines denote the 95% confidence intervals for the linear model fit. The different colors denote the different consumption sustainability ($FMI_C$) derivations. The black line shows the hypothetical direct 1:1 relationship between the two variables for comparison.
(PDF)

**S4 Fig.** Aquaculture exclusion analysis: The proportion of global exports (a) and imports (b) from 2012–2017 for each country analyzed arranged by FMIP. The size and color of the points shows the number of trade partners for each country (note difference in scale of y-axis). The top five exporters and importers are labeled in each panel.
(PDF)

**S5 Fig.** Aquaculture exclusion analysis: Figure depicting (a) imports to the United States of America (USA) from their top 25 import partners and (b) exports from China to their top 25 export partners from 2012–2017. In the maps, China and the USA are shaded based on their own production sustainability. The lines and points show the origin of imports to the USA (a) and destination of exports from China (b), respectively, with the color denoting the production sustainability of the trade partner. The thickness and darkness of the lines and points are scaled to the magnitude of the trade going to/from each country. The bar plots depict the proportion of total USA imports or Chinese exports attributable to each of top 25 trade partners,

respectively. The USA's top 25 import partners account for 94.34% of total imports to the USA from 2012–2017. China's top 25 export partners account for 89.90% of total exports from China from 2012–2017. The bars are shaded based on the trade partner's production sustainability.
(PDF)

## Acknowledgments

We thank Camille Kohler, Brian Garber-Yonts, Ben Fissel, and Mike Dalton for their insight and support in accessing the GTA data.

## Author Contributions

**Conceptualization:** Kayla M. Blincow, Alan C. Haynie, Brice X. Semmens.

**Data curation:** Kayla M. Blincow, Alan C. Haynie.

**Formal analysis:** Kayla M. Blincow.

**Funding acquisition:** Kayla M. Blincow.

**Investigation:** Kayla M. Blincow, Alan C. Haynie.

**Methodology:** Kayla M. Blincow, Alan C. Haynie, Brice X. Semmens.

**Project administration:** Kayla M. Blincow.

**Supervision:** Alan C. Haynie, Brice X. Semmens.

**Visualization:** Kayla M. Blincow.

**Writing – original draft:** Kayla M. Blincow.

**Writing – review & editing:** Kayla M. Blincow, Alan C. Haynie, Brice X. Semmens.

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
