## [Decision Letter · Decision Letter 0]

19 Jun 2024

PONE-D-24-10634Disparities between sustainability of country-level seafood production and consumption.PLOS ONE

Dear Dr. Blincow,

Thank you for submitting your manuscript to PLOS ONE. After careful consideration, we feel that it has merit but does not fully meet PLOS ONE’s publication criteria as it currently stands. Therefore, we invite you to submit a revised version of the manuscript that addresses the points raised during the review process.

**ACADEMIC EDITOR:**
Please ensure that your manuscript meets PLOS ONE's style requirements, including those for file naming. The PLOS ONE style templates can be found at https://journals.plos.org/plosone/s/submission-guidelines.In your Data Availability statement, you have not specified where the minimal data set underlying the results described in your manuscript can be found. PLOS defines a study's minimal data set as the underlying data used to reach the conclusions drawn in the manuscript and any additional data required to replicate the reported study findings in their entirety. All PLOS journals require that the minimal data set be made fully available. 

Upon re-submitting your revised manuscript, please upload your study’s minimal underlying data set as either Supporting Information files or to a stable, public repository and include the relevant URLs, DOIs, or accession numbers within your revised cover letter.

I agree with one reviewer that a weakness should be the assumption that sustainability was the same for both aquaculture and capture fisheries in a country. however, I am equally aware of the absence of specific information that could help the authors consider this aspect. I therefore also suggest that the authors try to use the commercial data of the taxonomic group to try to make a subdivision, albeit summary, into aquaculture and capture fishing. This could provide a further

---

## [Author Response · Author response to Decision Letter 0]

22 Oct 2024

Dear Editors,

We appreciate the thoughtful consideration of our work by the reviewers and have addressed their comments to the best of our ability. Our responses to each comment are provided below.

In particular, we expanded our analysis to include a duplication of our methods but using a dataset limited to trade relationships potentially associated with wild capture fisheries only. We present this parallel analysis to account for differences associated with aquaculture-produced products.

We believe our manuscript is greatly strengthened by the incorporation of the reviewer comments and is now ready for publication in PLoS ONE.

Thank you, 

Kayla Blincow

(on behalf of all authors)

ACADEMIC EDITOR:

1. Please ensure that your manuscript meets PLOS ONE's style requirements, including those for file naming. The PLOS ONE style templates can be found at https://journals.plos.org/plosone/s/submission-guidelines.

We have ensured that all files follow the journal’s style requirements, including file naming.

2. In your Data Availability statement, you have not specified where the minimal data set underlying the results described in your manuscript can be found. PLOS defines a study's minimal data set as the underlying data used to reach the conclusions drawn in the manuscript and any additional data required to replicate the reported study findings in their entirety. All PLOS journals require that the minimal data set be made fully available. 

We have updated our data, code, and documentation on Zenodo (DOI: 10.5281/zenodo.13799970)to make things more transparent. We include all necessary data files to run our analysis except for those associated with raw data from the listed third-party data source. Regarding the third-party data, we provided additional information on how to access those files. 

Upon re-submitting your revised manuscript, please upload your study’s minimal underlying data set as either Supporting Information files or to a stable, public repository and include the relevant URLs, DOIs, or accession numbers within your revised cover letter.

3. I agree with one reviewer that a weakness should be the assumption that sustainability was the same for both aquaculture and capture fisheries in a country. however, I am equally aware of the absence of specific information that could help the authors consider this aspect. I therefore also suggest that the authors try to use the commercial data of the taxonomic group to try to make a subdivision, albeit summary, into aquaculture and capture fishing. This could provide a further

We included an additional parallel analysis (called the aquaculture exclusion analysis in the update manuscript) addressing the issue of aquaculture more directly, duplicating our methods only removing trade relationships associated with potential aquaculture products (see L167–180). This parallel analysis is reported in the written results as well as in the supplement via figures and tables that mirror those generated for the core analysis (see Supplement).

Reviewer 1:

This is an interesting look at how the sustainability of seafood differs between countries, and between imports and exports. The authors seem to have a done thorough job of obtaining all available data and making well-reasoned assumptions to fill in missing information.

We appreciate the positive feedback from the reviewer.

Introduction

Line 52: what is “fisheries-induced evolution”?

We added some additional explanation of the impacts that arise from fisheries-induced evolution (L44-45).

Are there any data on sales of sustainably certified seafood products? Either globally or worldwide? It would be interesting to see this as a measure of consumer demand and willingness to pay for sustainable fisheries. Because that in turn can lead to more sustainable fisheries, even in the absence of a top-down push for greater sustainability.

There is work done in specific fisheries (e.g. Andersson and Hammarlun (2023) Ecological Economics), but we are not aware of data that includes this information on a scale that matches our analysis in this manuscript. The Marine Stewardship Council, the primary global eco-certification organization, actively supports research to explore this, but it has been challenging to demonstrate the impacts (e.g. Arton et al. (2020) Environmental Evidence). We discuss the roles of consumer-driven demand in the discussion section more thoroughly in light of the findings of our analysis (L496-523).

Is there any relevant international legislation around the sustainability of seafood to reference? Or oceans protection, since those two are often intertwined.

We added a section addressing the role of international agreements around seafood sustainability and the importance of national level implementation of these agreements (L47-56).

Has there been any similar research?

We cited similar research throughout the introduction and discussion, but we have now added an additional sentence explicitly listing previous studies as examples of research addressing global and national seafood sustainability (L91).

Methods

Section 2.2. was a bit difficult for me to follow at times since it was so heavy on equations. Not sure if there is anything that can be done differently, except maybe providing an example? One seafood product type for one country example might be useful for me, but other reviewers may not feel the need.

Previous drafts included an example, but it made the methods come across as disjointed. We ultimately decided relying solely on equations is a more transparent and direct way to describe the methods. 

Results

Are there certain types of seafood products that are driving some of these trends? E.g., are imports to the United States dominated by a couple of types of seafood products that the US doesn’t really produce domestically? This would be interesting to see and perhaps identify areas for improvement/potential projects.

Certainly, some countries tend to import more of certain types of products than others depending on the market that exists within each country. However, this goes beyond the scope of our analysis, as we were more interested in looking at global disparities in production and consumption seafood sustainability on a broad scale, rather than tracking trends in individual product types. That would be a substantially different analysis; one that we have explored writing up as a separate manuscript. 

Discussion

The key finding that countries producing more sustainable seafood end up consuming a lot of less sustainable seafood is quite interesting (lines 330-331) and I think merits more discussion. For example. in many developing countries small-scale fisheries at the local level account for a lot of the seafood consumption, so the country does not need to import a lot of fresh seafood to meet demands (although that is changing in many cities).

We added discussion of the implications of developing countries relying on their own production for seafood consumption (L432-438).

Lines 359-364: Any potential explanations why the sustainability disparity for the US is so much larger than similar countries? Why is so much of the import from those countries towards the bottom of the list?

We thank the reviewer for drawing attention to this point. We added discussion of the causes of the differences between the US and other high production sustainability countries (L453-458).

Why has there been decreased fisheries production? Overfishing, regulatory restrictions, etc…

Previously published literature suggests that the decline in production from developed countries in recent decades is associated with more stringent management and regulations (Ye and Gutierrez 2017). The same paper offers this as an explanation for why we see increased trade flows from developing to developed countries. We expanded discussion of these relationships based on comments from both reviewers (L423-426).

Lines 450-451: I would be curious about some statistics around this as I was under the impression that most fisheries were not sustainable. Or at least not certified sustainable.

We added a citation to this statement so readers can reference previously published analyses on sustainable seafood production in the developing world (L539).

Another thought/question:

Why does China dominate the export market? Is it mainly due to size (of population and coastline/ocean territory)? I also find it surprising that China’s sustainability is so low, given its economy. I would think it has the means to have greater oversight, but maybe it chooses not to?

We added language describing some potential reasons as to why China dominates the seafood export market (L62-463). These comments are in addition to the existing explanation given as to why sustainability is low—that the inability of consumers to distinguish between sustainable and unsustainable products results in a failure to economically incentivize large-scale producers to create sustainable products (L509-523). In general, we have tried to avoid painting China in a wholly negative light. More potential mechanistic explanations are available in Melnychuk et al. (2017), to which we point readers.

Reviewer 2 Comments:

This paper uses an index of fisheries sustainability by country and fish trade data to demonstrate that many countries that tend to manage their fisheries in a more sustainable fashion import substantial portions of their fish for consumption from countries that have lower sustainability standards. This is not particularly surprising, given that even if trade is random, countries with higher sustainability standards would be expected to on average import from countries with lower sustainability standards, although it could be that high sustainability countries trade with each other and low sustainability countries trade with other low sustainability countries, but this does not appear to be the case.

The authors also highlight the difficulty in tracing seafood source when reprocessing occurs, thus making true tracing of sustainability more uncertain.

Providing evidence for these important issues is valuable. The paper is well written and the methods seem clear.

We thank the reviewer for their cogent summary of our manuscript and their assessment of our research as valuable and well-written.

The major weakness/concern I have is the assumption that the sustainability of aquaculture in a country was the same as their capture fisheries. This needs to be explored perhaps by looking at the fraction of aquaculture production that is ASC or BAP certified by country. However, since close to 80% of aquaculture production comes from countries with low FMI scores I would guess that other than farmed salmon, almost all aquaculture products would have low scores.

Sustainability issues in aquaculture are quite different from capture fisheries so I believe a little more exploration by the authors of whether national FMI scores really reflect aquaculture sustainability is warranted.

While the trade data do not show whether the product is wild caught or from aquaculture, I suspect that the trade data on taxonomic group would be able to largely reflect the means of production. For example I suspect that almost all carp, catfish and shrimp that are traded come from aquaculture, as is probably true for most molluscs. The authors could compare global capture fisheries production to aquaculture production at the taxonomic level available in the trade data in order to make an estimate of the proportion of imports for each country that come from aquaculture. The key issue is whether the low sustainability score of imports for high FMIp countries is largely driven by their source being aquaculture.

We appreciate the reviewer highlighting the potential issues with combining aquaculture and wild capture products in our analysis. While these issues are not directly solvable given the limitations of the available data, the comments of the reviewer led us to conduct a parallel analysis that attempts to partially address the concern. In the parallel analysis (called the aquaculture exclusion analysis in the updated manuscript), we removed trade relationships associated with potential aquaculture products (see L167-180) and report the corresponding results alongside our original analysis (see Supplement). The results of the aquaculture exclusion analysis align closely with the results of our original analysis, suggesting that our findings are largely robust to the aquaculture assumptions we have implemented.

A closely related issue that the authors did not discuss is whether by achieving higher sustainability standards the best performing countries actually reduce their seafood production, thus leading to more imports with the net result that the overall environmental impact is negative. For example in the U.S. severe restrictions on several swordfish fisheries to reduce bycatch had the result that the U.S. imported more swordfish from countries with less concern about bycatch. The net effect was to reduce the sustainability of the swordfish being consumed. Is there evidence that high FMI countries have been reducing their wild capture consumption, while importing more seafood with lower FMI scores?

This is an important dynamic to the interaction of domestic production, although we believe it is outside the scope of our analysis given its complexity. At the global scale it would be challenging to identify this type of relationship, especially given the coarse nature of the FAO food balance sheet data (that is needed to estimate consumption). However, the GTA data would be a great option for doing a species-specific analysis of trade flows that could be matched with a different data source that characterizes consumption. Of course, the temporal dimension of the effect that you identify is important; we would expect sustainability measures to potentially lead to higher production in the long term, so this would make identification of the effect extremely challenging via analysis with annual comparisons. Previously published literature suggests that the decline in production from developed countries in recent decades is associated with more stringent management and regulations (Ye and Gutierrez 2017). The same paper offers this as an explanation for why we see increased trade flows from developing to developed countries. We expanded discussion of these relationships based on comments from both reviewers (L423-426).

The global trade data are available by subscription, but it should be possible for the authors to present a summary of the data in tabular form as they compiled it. I don’t know enough about the database to know what the options are.

We have provided a summary of the data for readers to review as well as additional information on where/how readers can obtain the GTA data in our updated data sharing documentation.

---

## [Editor Report · Decision Letter 1]

1 Nov 2024

Disparities between sustainability of country-level seafood production and consumption.

PONE-D-24-10634R1

Dear Dr. Blincow,

We’re pleased to inform you that your manuscript has been judged scientifically suitable for publication and will be formally accepted for publication once it meets all outstanding technical requirements.

Kind regards,

Rosa Luisa Ambrosio

Academic Editor

PLOS ONE

---

## [Editor Report · Acceptance letter]

13 Nov 2024

PONE-D-24-10634R1 

PLOS ONE

Dear Dr. Blincow, 

I'm pleased to inform you that your manuscript has been deemed suitable for publication in PLOS ONE. Congratulations! Your manuscript is now being handed over to our production team.

Kind regards, 

on behalf of

Dr. Rosa Luisa Ambrosio 

Academic Editor

PLOS ONE